# Evaluation of Point-of-Care Testing in Pharmacy to Inform Policy Writing by the New Brunswick College of Pharmacists

**DOI:** 10.3390/pharmacy10060159

**Published:** 2022-11-26

**Authors:** Lauren Hutchings, Anastasia Shiamptanis

**Affiliations:** New Brunswick College of Pharmacists, 686 St George Blvd, Moncton, NB E1E 2C6, Canada

**Keywords:** point-of-care testing, scope of practice, pharmacy services, policy

## Abstract

Pharmacy practice continues to advance, allowing professionals to contribute further to patient care and the healthcare system. Pharmacists are authorized to perform point-of-care testing (POCT) in seven out of ten Canadian provinces. In considering the potential for enhanced clinical decision-making with the opportunity to gain patient data at the site of care, the New Brunswick College of Pharmacists (NBCP) proceeded to draft regulatory amendments and a policy to enable POCT scope in New Brunswick. Policy writing is a core function of Provincial Regulatory Authorities in Canada as the process determines principles that direct pharmacy practice. Each province has a differing scope of practice and method for developing documents. This paper highlights the approach, analysis, and findings of the NBCP pursuant to drafting a POCT policy. The policy development process included a literature search and environmental scan of the ten Canadian provincial regulatory authorities along with other countries. The findings highlighted in this paper describe the use of POCT, quality assurance, regulatory framework, educational opportunities, and the role of pharmacy technicians in relation to POCT in a pharmacy setting. The approach NBCP took to engage professionals and decisions on the direction of the policy are described. As point-of-care services continue to expand in pharmacies, the insights by the NBCP can be utilized by other regulatory bodies or pharmacy professionals who are implementing or enhancing POCT policies or procedures within their organizations.

## 1. Introduction

Point-of-care testing (POCT) can allow access to clinically significant values or screening results for efficient care delivery, reducing accessibility concerns and wait times for laboratory data. In addition, data from such tests can assist pharmacists in fulfilling their duty to promote health and medication management. POCT is used in many healthcare settings, such as physician offices, nursing facilities, hospitals, and paramedic services [1]. Various uses of POCT include medication monitoring, chronic disease management and screening for some communicable infections. Within a pharmacy setting, screening for infection would require collaboration with public health and other stakeholders to establish a pathway for patients. Common specimens collected for these tests include nasal, throat, or buccal swabs and capillary blood samples. The World Health Organization describes POCT as representing six items: affordable, sensitive, specific, user-friendly, rapid & robust, equipment-free, and deliverable with additional criteria of real-time connectivity, ease of specimen collection, and environmentally sustainable proposed [2].

Canada does not nationally have a homogeneous scope of practice across each province. Provincial regulatory authorities (PRAs), also known as Colleges, collaborate with provincial governmental bodies to derive regulations that meet the province’s needs. The New Brunswick College of Pharmacists (NBCP) regulate all pharmacists, pharmacy technicians, community, and hospital pharmacies in the province. Regulatory amendments for POCT, which were approved in June 2022, focused on being more outcomes-based, including details regarding collaboration with other providers, referring where appropriate, and documentation requirements [3]. The Pharmacy Regulations in New Brunswick enable the administration of POCT subject to Council (Board) guidance. Therefore, POCT is not implemented in New Brunswick until the publishing of a guidance document. Pharmacy technicians are regulated providers under the College and are listed in the regulations as professionals able to administer tests under pharmacist supervision [3]. 

With the regulations in place, establishing a directional policy specific to POCT would be used along with the Code of Ethics, Standards of Practice, and applicable legislation to provide guidance and agency to set clear expectations. The National Association of Pharmacy Regulatory Authorities (NAPRA) published “Model Standards of Practice for Pharmacists and Pharmacy Technicians in Canada,” which has been adopted by most provincial Colleges, including New Brunswick. The NAPRA Standards include expectations for performing POCT that would be required to be met alongside the policy put forth by the College [4]. For pharmacists, the expectations include; determining POCT suitability, gathering informed consent, administering the test in a suitable environment, and interpreting results. Pharmacy technicians are expected to gather, administer, and document assessments and results of POCT [4]. Details regarding patient confidentiality and privacy are outlined within the NAPRA Standards and would apply in the context of POCT services. With practice standards specific to POCT articulated, the NBCP sought to consider aspects of POCT that would help to set parameters to support patient safety. It is important to analyze the various POCTs that pharmacy professionals utilize in other regions to understand how New Brunswick pharmacy professionals may implement them into their practice and the impact on patient care. This paper will explore NBCP’s evaluation of POCT in pharmacies and the direction taken by other jurisdictions to inform policy writing. The policy direction will be described based on presentation of findings to the Colleges Professional Practice Committee (PPC) and Council for approval.

## 2. NBCP Methods

The NBCP compiled research evidence and findings to support decision making.

### 2.1. Literature Review

The main objectives of the literature review were to describe the use of POCT and safety considerations for POCT in pharmacy practice. A systematic search of PubMed using a search strategy as follows: (“Pharmaci*” OR ”Pharmacy”) AND (“Point of Care test” OR “POCT” OR “Rapid test”). Articles were limited to the English language and to a time period of 10 years from 16 May 2022. Studies were excluded based on the title and abstract if a pharmacist did not perform POCT in a community pharmacy setting, if the device required laboratory processing, and articles revolving around the SARS-2 virus (COVID) as guidance on COVID testing existed in New Brunswick under emergency measures. Within the relevant articles, a review of reference lists was performed to obtain additional papers.

### 2.2. Environmental Scan

An environmental scan was completed of the nine other Canadian provinces. The regulations regarding POCT were investigated along with applicable policies through Provincial Regulatory Authority websites. PRA’s were contacted for clarification on their scope of practice as required. The United States, along with other countries frequently appearing in the literature, such as Australia and Ireland, were examined to understand the framework used for regulating POCT.

### 2.3. Professional and Stakeholder Engagement

Pharmacy professionals were consulted through a survey to understand what tests are valued and their comfortability with implementing POCT. New Brunswick health regulators were consulted throughout this process. Governmental bodies including Public Health were engaged. Two PPC meetings were held to set a direction for the policy and review the draft policy. Following approval of the draft, the policy was presented to Council for approval.

## 3. Usage of POCT in Community Pharmacy

POCTs are commonly used for chronic disease management and infectious disease screening. Within the chronic disease management, Hgb A1C, lipids, renal function, and international normalized ratio (INR) tests will be explored. For infectious disease screening, Streptococcal, influenza, HIV, hepatitis C (HCV), and COVID-19. 

### 3.1. Chronic Disease Management

Pharmacists access to values such as Hgb A1C, lipids, and renal function through POCT can enhance clinical decision-making. Health Canada licenced devices include handheld instruments that provide an Hgb A1C reading within 5 min with a small blood sample [5,6]. A review of pharmacy-provided Hgb A1C POCT included two observational studies that found no significant difference between the patient’s baseline and follow-up Hgb A1C [7]. One of the observational studies took place in Alberta and had pharmacists use POCT to assist in prescribing insulin for type 2 diabetic patients [8]. Following 26 weeks of care, there was an Hgb A1C reduction of 1.8% (95% CI 1.4–2%), showing comparable results to physician-led interventions [8].

Lipid tests licenced in Canada tend to entail a full panel, including total cholesterol, HDL, LDL, and triglycerides [6]. A review of three studies involving pharmacists’ completed lipid testing showed no significant difference in total cholesterol at six months compared to baseline or physician mediated care [7]. Two observational studies showed a non-significant reduction in triglycerides by 21.68 mg/dL following two years (95% CI 34.74 to 8.61 mg/dL) [7]. Another trial took place in Alberta and Saskatchewan pharmacies [9]. Patients were randomized to pharmacist intervention where education, POCT, referral to a physician and regular follow-up occurred versus general counselling with minimal follow-up (*n* = 675). The primary composite endpoint involved a physician completing a cholesterol panel or a prescription or dose change of a cholesterol-lowering medication. This endpoint occurred in 57% of the intervention patients versus 31% in the general counselling group (OR 3.0 95% CI 2.2–4.1) [9]. The results showed that pharmacist POCT could enhance cholesterol risk management.

Pharmacists are key in medication dosing based on renal function and managing chronic kidney disease. A pilot study in Ontario used POCT to screen adults at risk for chronic kidney disease [10]. Of the 89 participants, 11.24% (*n* = 10) were found to have an eGFR <60 mL/min/1.73 m^2^ leading to referral [10]. A study in the Netherlands used a clinical decision support system to create alerts for seven antibiotics where eGFR values were unavailable in patients over 70 years old [11]. A small proportion of alerts (2.2%) led to an initiation of a POCT totaling 1988 tests completed. Overall, 15 prescriptions were modified based on the renal function test result [11]. Pharmacists are positioned in the community to investigate renal impairment and act on results to prevent medication accumulation. Depending on the scope of practice in the jurisdiction, action may be taken through a pharmacist adaptation, or a referral may be required to another healthcare provider. 

Patients on the anticoagulant warfarin require INR testing and dose adjustments based on the results. In four observational studies using INR POCT, pharmacists either worked collaboratively with other providers or led the warfarin management [7]. A pooled analysis demonstrated a 7.99% mean difference in time in therapeutic range (TTR) for pharmacist care versus physician-led care, which was not significantly different (95% CI −0.74–16.71%) [7]. New Zealand pharmacists led an anticoagulation clinic for 693 patients with atrial fibrillation resulting in a mean TTR of 78.6% (95% CI 49.3–100%) [12]. A subgroup analysis compared pharmacy testing to physician care showing an increase in TTR by 16.7% (*p* < 0.001) [12]. Although an improvement in patients monitoring values following pharmacist administered POCT was not seen in all studies; increased accessibility was not considered. Community pharmacist POCT removed the need for patients to receive a laboratory requisition, attend a facility for sample collection, and wait for results.

### 3.2. Infectious Disease POCT

Infectious disease screening through POCT can identify a pathogen or confirm an antibiotic indication, thus strengthening stewardship [13]. Tests in this category include; streptococcal, influenza, HIV, HCV, and COVID-19. A Canadian streptococcal retrospective study was completed across British Columbia, Alberta, and Nova Scotia pharmacies [14]. Patients with associated symptoms were tested (*n* = 7050), leading to a positivity rate of 25.4%. Out of the 1795 positive patients, 1234 received a same-day prescription. In a follow-up patient survey, 91% of respondents identified the pharmacy as convenient and 79% perceived value [14]. This trial excluded children under five years old. However, it is essential to note that due to the sensitivity being below 100% and the increased risk of complications in children, a throat culture should be followed up in the case of a negative pediatric test [15]. Throat cultures may not be accessible to pharmacists in the absence of lab testing, therefore referral may be warranted in such a case. It is important to use assessment tools as it is cited that the asymptomatic carriage rate of streptococcus is 7.5%, and tests cannot differentiate between infection and carrier status [13]. Prior assessment of symptoms and physical features can optimize the predictive values of POCTs. When pharmacists perform testing there is an expectation that they have the knowledge, skills, and judgement in assessing the patients features that determine whether the test is indicated. Implementation of these services in some cases involved input from advisory boards, training sessions, and informing nearby general practitioners that these services were being implemented [16].

Influenza POCT typically involves a sample from the nasal cavity or nasopharynx, with most devices identifying types A and B or, in some cases, only one type [17]. A study in United States pharmacies led to 22.9% (*n* = 19) of patients testing positive and fifteen being prescribed oseltamivir [18]. An observational study at two independent pharmacies in the United States (*n* = 73) led to thirteen positive influenza cases and twelve patients receiving anti-viral prescriptions [19]. A report reviewed an influenza POCT protocol using emergency department medical records to determine how many patients would be eligible for a pharmacy test based on pre-set criteria [20]. The protocol excluded patients from a test based on the patient’s pulse, blood pressure, respiratory rate, temperature, oxygen saturation, alertness, the timing of symptoms or diagnosis of a disease that can be immunocompromising. Out of 451 patients investigated, 34% would have been eligible for an influenza test. It was concluded that proper protocols could ensure tests are not misused but must not be too restrictive in preventing care [20].

Sexually transmitted infections such as HIV and HCV can be detected through antibody POCT, although the test value is limited by antibody production time post-infection. The United States Centers for Disease Control recommends a 90-day window for antibody POCT [21]. Thus, tests should be used as a screening tool in at-risk populations instead of following an exposure. With reports stating that 50% of transmissions are unaware, screening in an accessible setting can link patients to care earlier in infection [22]. A pharmacy serving the Washington homeless population tested ten walk-in patients resulting in eight positive HCV tests and no positive HIV results [21]. The APPROACH study in Alberta and Newfoundland and Labrador undertook HIV POCT [22]. The participating pharmacists completed training provided through the study, which included a module, in-person session, and assessment. In this 6-month pilot, one case was detected (*n* = 123). In addition, a questionnaire was provided to patients displaying patient comfortability at the pharmacy and noted the likelihood of following the counselling advice [22]. Within a Walgreens trial in the United States (*n* = 3630), 0.8% of HIV tests were reactive, with four of those patients later lost to follow-up [23]. HIV and HCV are listed on the national notifiable disease list, requiring public health involvement [24]. Engagement with local public health units is vital for policy development to ensure pharmacy services fit within the broader health system.

During the COVID-19 pandemic, many antigen POCTs became available, allowing access to quick results to mitigate the further spread of the virus. Health Canada authorized which personnel (health professional or individual) may administer each test and differentiates between lab, point-of-care or home setting [25]. The pandemic has increased the use of throat swabs which will likely make POCT more commonplace for other infectious diseases [13]. Beyond infectious diseases, POCT consists of various tests beyond the ones discussed above. Other tests include blood gases, electrolytes, ketones, albumin, medication analytes, fecal occult blood, helicobacter pylori, Lyme antibodies, and celiac disease tests [6,26]. These tests were not observed as frequently in the pharmacy setting within the literature and thus were not explored in depth within this paper. Overall, POCT in community pharmacies typically involves a non-invasive specimen collection process and a device that provides a result in a short time frame, allowing immediate care. POCT in pharmacies connects patients to screening services to improve chronic condition management or infection detection. 

## 4. Quality Assurance

Quality assurance or control in the context of POCT is a term used to ensure the performance of devices is acceptable [27]. Such measures should follow the manufacturer’s instructions. The Australian Department of Health provides a detailed description of quality control requirements for POCT that conveys provider responsibility in maintaining patient safety [27]. External quality assurance involves a sample where the expected result is known. Whether the test is qualitative or quantitative, the result should concord with the known value. The referenced requirements note the frequency of testing should occur for each new batch or lot of reagent or cartridge, if the manufacturer does not specify a timeframe [27]. Samples for external testing may be available through the manufacturer or an external source. In the absence of such a test, sample parallel testing with an accredited laboratory may be considered. Internal controls include control solutions and calibration. Devices have internal controls, including ensuring sufficient specimen quantity to ensure satisfactory device operation [27]. Within the policy drafted by NBCP, professionals are expected to perform quality assurance such as calibration according to manufacturer specifications. The quality assurance steps would be required to be documented within a Standard Operating Procedure. In the case that the device is not functioning to required standards it should not be utilized.

CADTH reviewed the accuracy of three infectious disease POCTs used for patient self-testing or pharmacist testing [28]. When POCT for HCV, group a streptococcus, and influenza A and B were compared to laboratory values, the POCT performed reliably well for diagnosing such infections. It should be noted that influenza A and B tests have a low sensitivity therefore, a conventional laboratory test should follow up on a negative test [28]. In a study out of eight pharmacies in Italy, patients (*n* = 106) completed POCT while simultaneously having a sample drawn by a physician for laboratory testing [29]. The POCT values for glucose, cholesterol, HDL-cholesterol, creatinine, uric acid, AST, and ALT were satisfactorily correlated (slopes ranged between 1.23 AST and 0.92 uric acid). However, triglyceride values were significantly different between POCT and laboratory results with the POCT median being higher (1.627 mmol/L vs. 0.950 mmol/L). The triglyceride difference may be attributed to improper fingertip cleaning, as glycerin-containing hand creams and soaps could impact instrument readings [29].

## 5. Regional Variation in POCT

### 5.1. POCT across Canada

Canada does not have national POCT requirements [30]. The Canadian Standards Association has adopted the International Organization of Standardization guidance specific to POCT (ISO22870), which includes hospital, clinic, or ambulatory care settings [30]. Each province has varying scopes and approaches to policy development regarding POCT. Table 1 summarizes the provincial scope of practices and applicable policies. When COVID-19 testing is excluded from the POCT category, of the ten PRAs, seven have enabled POCT implementation. The COVID-19 pandemic compelled provinces to make amendments to allow for the administration of COVID-19 testing. The Medical Laboratory Licensing Act governs Saskatchewan POCT regulations [31]. As such, pharmacists can only provide POCT if authorized through a medical laboratory license or in the case of an emergency enactment. This was the case for COVID-19 testing, which has now been rescinded. Therefore, pharmacists are currently not authorized to perform tests, and doing so under a laboratory licence is not commonplace [31,32].

Manitoba similarly authorized pharmacists to administer COVID-19 tests, however for other tests, pharmacists can only interpret patient-administered tests [33]. Ontario expanded POCT beyond COVID-19 on 1st July 2022, for medication management for chronic diseases [34]. Testing is limited to glucose, Hgb A1C, lipids, and INR [34]. Similar restrictions exist in Prince Edward Island (PEI), where pharmacists or pharmacy technicians may administer the same tests, excluding lipids [35]. In summary, Manitoba allows COVID-19 tests in pharmacies, whereas Ontario and PEI limit tests to a prescribed list. In British Columbia, POCT is not within or excluded from the scope of practice of pharmacy professionals, however, professionals may require authorization through the Diagnostic Accreditation Program managed by the College of Physicians and Surgeons of British Columbia [36,37]. Provinces including Quebec, Newfoundland and Labrador, Nova Scotia, and Alberta do not restrict the type of test but allow open access [38,39,40,41].

In provinces where policy has been developed, common themes are shared. The PRA for British Columbia references the provincial Ministry of Health, which provides an approach generalized to any community-based health provided POCT service [42]. Newfoundland and Labrador Pharmacy Board provide professionals with guidelines for POCT, whereas Nova Scotia includes a POCT section within their testing standards of practice document [39,40]. Alberta was the forerunner for pharmacists providing testing services and has a standard of practice, guidance, resource list, and a paper on considerations when implementing POCT [41,43,44,45,46]. An important recurrent theme within these documents is for professionals to understand devices’ accuracy, precision, and reliability. Pharmacists must use clinical decision-making to decide if the test is appropriate for the intended purpose and if a laboratory test may be more suitable. Duplication of tests is noted in policies to ensure healthcare resources are not used unnecessarily. Trained staff should be familiar with the pharmacies standard operating procedure should be established, including instructions of use to allow the uniform function of devices and a record of quality control measures being maintained [31]. The testing environment should allow for privacy and infection control measures. 

**Table 1 pharmacy-10-00159-t001:** POCT Scope of Practice Across Canada.

Province	POCT Scope	Policies
British Columbia	Not explicitly within, nor excluded from scope. Pharmacists and pharmacy technicians may perform tests [36].	The provincial Ministry of Health created a policy to guide implementation [42].
Alberta	Within scope for pharmacists and pharmacy technicians [41].	Standards of Practice and guidance document along with resource recommendations is available [41,43,44,45,46].
Saskatchewan	Pharmacists may only perform POCT under a laboratory licence or if it is enabled by Registrar in extraordinary circumstances [31].	Policy on laboratory tests includes a section on POCT in the event it is enacted by the Registrar [31].
Manitoba	Pharmacists can administer COVID-19 POCT tests and interpret patient administered tests per the Pharmaceutical Act [33].	No applicable policies.
Ontario	Within scope. Pharmacists, interns, students, and technicians are enabled to administer. Limited to Hgb A1C, blood glucose, lipids, and INR [34].	Guidance document on expectations when piercing the dermis for POCT [34].
Quebec	Within scope for pharmacists [38].	No applicable policies.
Nova Scotia	Within scope for pharmacists, technicians, interns, and students [40].	The testing standards of practice document covers POCT [40].
Prince Edward Island	Within scope for pharmacists. Limited to Hgb A1C, blood glucose and INR as per the Regulated Health Professions Act Pharmacist and Pharmacy Technician Regulations [35].	No applicable policies.
Newfoundland and Labrador	Within scope for pharmacists [39].	Guidance document is available [39].

### 5.2. POCT Outside of Canada

In the United States, the federal Food and Drug Administration (FDA) approves tests that can be completed by health professionals who possess a Clinical Laboratory Improvement Amendments application or CLIA waiver [1]. Pharmacists must apply for a waiver through their State Agency, although state laws can restrict pharmacists from applying [1]. Other state regulations set limitations on the purpose of testing or make requirements that make it challenging to obtain a certificate [1,47]. Tests are listed on the FDA website and are considered to be low risk and intuitive to perform while recognizing errors still exist and can have determinantal outcomes [26]. There are over 120 analytes approved, including endocrinology, urinalysis, hematology, virology, bacteriology, immunology, toxicology, and general chemistry categories [26]. The percentage of pharmacies that have a waiver differs from the highest in Alaska with 60% to Nevada with 0% of pharmacies [1]. Collaborative practice agreements are available in 49 states allowing pharmacist prescribing and action to be taken based on results [47]. In regulation changes, states such as Idaho allow pharmacist prescribing based on a POCT results for influenza and streptococcus [48]. The pandemic altered the trajectory of pharmacy services in the United States as the number of waivers issued increased, and the recognition of pharmacists as providers allowed a pathway for third-party reimbursement [49]. As of 8 April 2020, pharmacists were able to administer approved COVID-19 POCTs, overriding any state or local restrictions on pharmacists obtaining CLIA-waivers [49]. Although, the ability to provide tests ceases when the executive orders expire with the resolution of the public health emergency [49].

Pharmacists are able to perform point-of-care testing in other countries, including Australia, the United Kingdom, Ireland, and New Zealand [13,27,50]. Australian pharmacists follow the regulatory framework set forth by the National Pathology Accreditation Advisory Council, which has a guideline document specific to POCT requirements based on standards from the International Organization for Standardization (ISO-22870) [27]. Australia’s POCT guidelines indicate the importance of reviewing manufacturer instructions in detail as using devices outside of such directions would be off-label [27]. Each pharmacy would require a designated regulated practitioner to direct the operations of the POCT services and create organization-specific policies to manage risk. Training, environment, steps taken at each testing phase, and workplace safety are included. Ireland pharmacists can provide POCT to facilitate monitoring and screening [50]. The Pharmaceutical Society of Ireland provides a document on pharmacy staff training, testing equipment management, and quality assurance measures [50]. In addition, pharmacists must comply with the Health Products Regulatory Authority guidelines, which cover POCT in community care [51]. Similar to New Brunswick Regulations, where devices must be Health Canada approved, devices in Ireland must achieve performance criteria established by the Irish Medicines Board [3,50]. 

## 6. Pharmacy Professional Competency Considerations

### 6.1. Education

Pharmacy schools undergo accreditation through the Canadian Council for Accreditation of Pharmacy Programs (CCAPP). POCT is an example of developing practice skills in the 2018 CCAPP standards under criterion 4.2 [52]. The accreditation standards for pharmacy technician programs do not directly allude to POCT administration [53]. When students move into practice, the competencies are outlined in NAPRA’s entry to practice document [54]. Pharmacists must have the skills to interpret laboratory and diagnostic data and assess technology safety and efficacy. 

An environmental scan by the Canadian Agency for Drugs and Technologies in Health (CADTH) in 2017 noted that there is no standardized training or certification available for health professionals [30]. Although this scan was not focused on the pharmacy setting, training programs reported were institution or health authority specific. A recent continuing education course by a professional development provider was created, providing online modules covering how to perform glucose, Hgb A1C, lipid, and INR POCT [55]. CADTH consulted with clinical laboratory personnel who highlighted that device-specific training is more beneficial than generalized training sessions [30]. The United States National Academy of Clinical Biochemistry (NACB) defines POCT as a clinical laboratory test by clinical personnel whose primary training is not in the clinical laboratory sciences or tests administered by patients [5]. Due to operators not being laboratory-trained, the NACB strongly recommends training programs that cover pre-analytical through to post-analytical stages [5]. In the United States, a 20 h certificate program is available through the National Alliance of State Pharmacy [15]. This program is cited as a resource recommended by the Alberta College [45]. An evaluation of the training program was completed through a survey of pharmacists in Arkansas [56]. Of the 23 respondents, 74% felt very or fully prepared to implement POCT [56]. Pharmacists reported that other resources would be helpful, including comparing manufacturers, patient intake forms, and pharmacy protocol examples.

### 6.2. Pharmacy Technicians Role in POCT

Pharmacy technicians are regulated and licensed by the NBCP. Pharmacy technicians must attend an accredited education program and have a defined scope of practice. Pharmacy technicians play a crucial role in the health care system, often responsible for many operational components of practice. Thus New Brunswick permits pharmacy technician participation in the technical components of POCT services [3]. Other PRAs allowing technician administration includes Ontario, Alberta, Nova Scotia, and British Columbia. It is relevant to note that various articles discuss the suitability of pharmacy technicians’ involvement in POCT. In the United States, pharmacy technicians requirement for registration depends on the state, in some states pharmacy technicians may be able to perform a POCT while having the pharmacist interpret the results [57,58]. Roles that are applicable to the pharmacy technician role in New Brunswick beyond administration includes inventory management of POCT supplies and expiration, scheduling appointments, consent forms, record keeping, and coordinating with other providers [57]. Technicians are required to be certified in Tennessee and Ohio where a survey was distributed to pharmacists. Respondents agreed that technicians should be involved (81%) and had no reservations about delegating POCT activities to technicians (84%) [59]. Another article evaluating the technician’s role saw a 16% difference in the number of tests completed compared to control pharmacies without technicians [60]. A survey of patients in Tennessee showed satisfaction with the services provided by the technician (94%) and believed the personnel showed professionalism (95%) [60].

## 7. POCT Endeavors Undertaken by NBCP

### 7.1. Pharmacy Professional and Stakeholder Engagement

As part of the policy development process, pharmacy professionals were engaged through a survey. The survey results were utilized in meetings with the PPC when determining the direction of the policy and in the drafting stages. The survey was completed by pharmacists, pharmacy managers, technicians, and pharmacy students. Respondents had the opportunity to select POCT categories they saw value in, benefits of the service, and barriers that would impact implementation. Competencies that professionals would need to review prior to administering POCT were explored, including confidence in creating a standard operating procedure, collecting specimens, assessing the analytical validity of the test, quality control measures and acting on the result. Using a Likert scale from 1 to 10, the average likelihood of implementation was selected. NBCP identified that professionals recognized medication management for chronic diseases as beneficial tests to implement in practice. In addition to professional engagement, governmental bodies including Public Health were consulted through the process.

### 7.2. New Brunswick POCT Policy

Following the completion of a literature review, environmental scan, and survey of professionals, the PPC met to set a direction for the policy based on the evidence presented. Based on discussions at the PPC meeting, the College moved forward with POCT for the purpose of medication management for chronic diseases, similar to Ontario and Prince Edward Island, although there will not be a defined list. Allowing for this testing purpose will enhance the pharmacist’s ability to provide patient care and take action on results through medication management. Testing for purposes such as infectious disease would require a structured pathway for Public Health notification. New Brunswick pharmacists are enabled to assess and prescribe for minor ailments, excluding conditions such as influenza or streptococcus in which POCT is available. Therefore, positive results of some infectious diseases would require referral to another provider in the absence of pharmacist prescribing. A systems-based approach will be taken through collaboration with Public Health to consider widening the purposes in the future to align with the government’s primary care plans and may include other purposes such as infectious disease screening. NBCP is working towards implementing the approved policy. The next step is to determine key performance indicators that would determine the utility of the policy in practice.

## 8. Conclusions

This analysis reviewed the usage of POCT services in pharmacy settings with the objective of using this information to develop policy that protects and promotes the health and well-being of New Brunswickers. Common themes were identified in other jurisdictional documents, which provide insights into key components that should be considered. Including a literature review, environmental scan, and engagement through a survey into the framework of drafting new policy initiatives aligns with the right-touch regulation model [61]. Following this standard encourages the policy approach to align with the amount of potential patient risk. This approach can apply to professionals implementing organizational-based procedures for POCT services. As pharmacy practice evolves, regulatory bodies must continue to systematically examine policies to allow safe streamlined care. 

## Data Availability

Not applicable.

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
