# Peer review of "Evaluation of Point-of-Care Testing in Pharmacy to Inform Policy Writing by the New Brunswick College of Pharmacists"

_pharmacy, 2022, doi:10.3390/pharmacy10060159_

Round 1

Reviewer 1 Report

The Communication article “Establishing a Point-of-Care Testing Policy: The New Brunswick College of Pharmacists Experience” by Lauren Hutchings and Anastasia Shiamptanis describes the use of POCT, quality assurance, regulatory framework, educational opportunities, and the role of pharmacy technicians in relation to POCT in a pharmacy setting in Canada. Moreover, the authors summarize results from studies on outcomes after POCT use.

General remark:

The study covers a relevant topic and is well written in most parts. I have several (minor) comments and some suggestions that may help to improve the manuscript.

Specific comments:

- The authors summarize the results of some studies on POCTs use in pharmacies. Was there a systematic literature with appropriate methodology performed to identify these studies? If not, there might be biases due to studies that were not found in the literature.

- For readers (especially for those not familiar with the Canadian regulations) it would helpful if the authors can clearly write, which POCTs are allowed to be performed in pharmacies. Is there a predefined list or can pharmacist perform any approved POCT?

- Moreover, for international readers it would be interesting to briefly mention the role of patients. In many other countries, patients with symptoms of illness would not go to pharmacies but would rather visit (outpatient) physicians. Are pharmacies in Canada established first contact points for medical services of patients with illness?

- I assume that pharmacists Canada are allowed to make clinical decisions after POCT use, such as prescription of drugs. In Germany and probably many other (western) countries, this would be unthinkable. From a German perspective, the role (and responsibilities) of pharmacists seems to be very different in Canada.  

- In cases where the authors describe study results, it would be helpful to briefly describe what “usual care” actually means.

- I am not sure about Table 2. There are much more studies than one study, that describe the diagnostic accuracy of POCTs for Influenza A and B. A quick literature check revealed that there are multiple systematic reviews on the test performance of POCTs for Influenza A and B.

- The authors state that assessment tools, such as the Centor score, can be used for assessment of symptoms and physical features prior POCT use (Strep A). This is a strong statement, since assessment of symptoms and physical features is usually performed by trained physicians. Is pharmacy personnel in Canada trained to assess symptoms and physical features? With other words, pharmacy personnel in Canada take on tasks that are usually performed by physicians in other countries. It would interesting to know more about the perspectives of physicians in Canada on POCT use in pharmacies. Are there any critical opinions from outpatient physicians?

- “External quality assurance involves a sample where the expected result is known. Whether the test is qualitative or quantitative, the result should concord with the known value. The referenced requirements note the frequency of testing should occur for each new batch or lot of reagent or cartridge, if the manufacturer does not specify a timeframe [27].”  Does this mean that pharmacists are required to perform these tests for external quality assurance? Are there independent authorities, which are responsible for this process? What happens if pharmacies fail these external quality assurance tests?

- Line 362: “A1C” should be HbA1C.

- Table 2: The authors may consider using the full names of the Canadian provinces or they might provide an abbreviation list.

Author Response

The Communication article “Establishing a Point-of-Care Testing Policy: The New Brunswick College of Pharmacists Experience” by Lauren Hutchings and Anastasia Shiamptanis describes the use of POCT, quality assurance, regulatory framework, educational opportunities, and the role of pharmacy technicians in relation to POCT in a pharmacy setting in Canada. Moreover, the authors summarize results from studies on outcomes after POCT use.

General remark:

The study covers a relevant topic and is well written in most parts. I have several (minor) comments and some suggestions that may help to improve the manuscript.

Specific comments:

- The authors summarize the results of some studies on POCTs use in pharmacies. Was there a systematic literature with appropriate methodology performed to identify these studies? If not, there might be biases due to studies that were not found in the literature.

Thank you for this feedback. We have added a methods section describing the literature review and environmental scan undertaken. 

- For readers (especially for those not familiar with the Canadian regulations) it would helpful if the authors can clearly write, which POCTs are allowed to be performed in pharmacies. Is there a predefined list or can pharmacist perform any approved POCT?

Within the introduction, details have been added to highlight how the scope of practice in Canada differs province by province. Table 1 describes the POCT-specific regulations in each province, including the list of tests approved in the provinces with limitations (Ontario & Prince Edward Island).    

- Moreover, for international readers it would be interesting to briefly mention the role of patients. In many other countries, patients with symptoms of illness would not go to pharmacies but would rather visit (outpatient) physicians. Are pharmacies in Canada established first contact points for medical services of patients with illness?

Within the introduction, there has been an addition to describe this point. POCT that screens for infectious diseases would require work with public health and other stakeholders to establish a pathway for patients. In section 10, we mention that POCT was decided to be limited to medication management as this pathway has yet to be established, and pharmacists could not take action such as prescribing in the event of a positive screening test.

- I assume that pharmacists Canada are allowed to make clinical decisions after POCT use, such as prescription of drugs. In Germany and probably many other (western) countries, this would be unthinkable. From a German perspective, the role (and responsibilities) of pharmacists seems to be very different in Canada.  

Prescribing scope of practice differs among provinces similar to POCT scope. Section 10 has been edited to make it clear that the direction of the policy to allow for POCT for chronic diseases was in part due to the limitation in prescribing function, as pharmacists may only prescribe for minor ailments which excludes conditions such as influenza and streptococcus that may require referral to another provider.

- In cases where the authors describe study results, it would be helpful to briefly describe what “usual care” actually means.

Agreed that “usual care” is a broad term. It has been defined in each instance it was used in section 3.

- I am not sure about Table 2. There are much more studies than one study, that describe the diagnostic accuracy of POCTs for Influenza A and B. A quick literature check revealed that there are multiple systematic reviews on the test performance of POCTs for Influenza A and B.

This table has been removed as it did focus on specific brands of POCT, in some cases they may not be available or applicable internationally.

- The authors state that assessment tools, such as the Centor score, can be used for assessment of symptoms and physical features prior POCT use (Strep A). This is a strong statement, since assessment of symptoms and physical features is usually performed by trained physicians. Is pharmacy personnel in Canada trained to assess symptoms and physical features? With other words, pharmacy personnel in Canada take on tasks that are usually performed by physicians in other countries. It would interesting to know more about the perspectives of physicians in Canada on POCT use in pharmacies. Are there any critical opinions from outpatient physicians?

Opinion pieces from physicians regarding pharmacist scope of practice have focused on pharmacists prescribing autonomously. Authors have not come across published opinions specific to pharmacists performing POCT. In provinces where testing for infectious disease is permitted, pharmacists are expected to self-assess to determine if they have the knowledge, skills and judgement to perform the assessment. A sentence was added to section 3.2, paragraph 1, to reiterate this point. We have decided to refrain from mentioning specific assessment tools.

- “External quality assurance involves a sample where the expected result is known. Whether the test is qualitative or quantitative, the result should concord with the known value. The referenced requirements note the frequency of testing should occur for each new batch or lot of reagent or cartridge, if the manufacturer does not specify a timeframe [27].”  Does this mean that pharmacists are required to perform these tests for external quality assurance? Are there independent authorities, which are responsible for this process? What happens if pharmacies fail these external quality assurance tests?

The following has been added to section 4, “Quality Assurance,” to inform the reader how NBCP has highlighted the expectations revolving around this process. “Within the policy drafted by NBCP, professionals are expected to perform quality assurance such as calibration according to manufacturer specifications. The quality assurance steps must be documented within a Standard Operating Procedure. If the device is not functioning to required standards, it should not be utilized as this would put patient safety at risk.”

- Line 362: “A1C” should be HbA1C.

Edited to HbA1C

- Table 2: The authors may consider using the full names of the Canadian provinces or they might provide an abbreviation list.

This has been edited in what is now Table 1.

Reviewer 2 Report

The paper is called “Establishing a Point-of-Care Testing Policy: The New Brunswick College of Pharmacists Experience”.

I expected to learn about the experience in implementing the point-of-care test (POCT) policy in New Brunswick: what are its features, why was it introduced, and which policy objectives should be reached (and which KPIs were defined for measuring progress), which difficulties in preparation and implementation were encountered, and which were the experiences of pharmacists, whether or not, and to which extent, the objectives were reached, possible hindering and enabling factors.

The abstract already suggested that “the New Brunswick College of Pharmacists Experience”, as stated in the title, might not be addressed. In the Introduction, recent regulatory amendments as of June 2022 were mentioned. I was wondering how an impact assessment were possible, after such a short time. But I was still expecting to learn about the preparation and implementation phase. But apart from a brief outlook in the Introduction, only section 8 informed about the policy development.

In the abstract, the New Brunswick College of Pharmacists (NBCP) Strategic Plan 2023 is mentioned. I was confused because this Strategic Plan points to the future, and this might not be the basis for the implementation of a measure whose impact were intended to evaluated.

The abstract states: “With the approval of the amendments, NBCP developed a policy to guide pharmacy professionals in implementing safe POCT services that provide quality patient care. As part of NBCP’s policy development process, a literature search and environmental scan of the ten Canadian provincial regulatory authorities along with other countries was completed.” I had to continue reading subsequent sections of the paper to understand that findings of this review were content in this paper. I will come back and comment on the objective of the paper below.

In the Introduction, second paragraph, the new regulations are presented, with an outlook on the roles of the different professionals in (community) pharmacy. Then it is written: “With minimum expectations set in place, the NBCP sought to consider aspects of POCT that would help to set parameters to support patient safety. This paper will explore the usage of POCT in community pharmacies, its impact on patient outcomes, and quality control measures.” I was wondering how the report of the implementation of the POCT policy in New Brunswick is already possible a few weeks after the regulatory change. The following sentence (“An analysis of provincial regulatory authorities, other countries, education perspectives, and the role of pharmacy technicians will provide an understanding of the POCT environment in pharmacy practice to inform policy development”) and the rest of the paper taught me that the measure is, at the time of the writing of the paper, being implemented, and no findings on experiences or learnings is to be expected.

Information on the methods is missing (apart from mentioning of the literature review and “environmental scan” in the abstract – see also below specific comments) though it might have helped to understand. This paper would benefit from a methods section.

The remainder of the paper is the presentation of a targeted review on some elements of POCT in community pharmacy globally and in Canada. However, it is not clear why certain topics were included and others not. Given the missing methods section, the readers do not know why some pieces of literature were included, and others were not. It rather reads as a technical report, or textbook, with some “self-marketing” for the pharmacists’ profession (see below).

I do not say that the information reported is not interesting. But I do not understand why this paper was written. If it aims to inform colleagues in the NBCP, other formats than a scientific paper (e.g. a booklet) might be more appropriate. There is no need to do a narrative review of systematic reviews in a paper.

I urge for a complete rewrite of the paper, upon clarification of its aim. Any misleading information on a possible reporting of experiences of the new policy should be avoided; thus the title needs to be changed. The authors need to define a research question. They can report some of the findings of the review, but they need to present the findings in light of the new POCT policy in New Brunswick: What does the international experience tell them? How did it feed their policy development? What are learnings to consider?

Specific comments:

The “New Brunswick College of Pharmacists” (NBCP): For readers not familiar with the Canadian pharmacy system, it would be helpful to provide further information on the NBCP. Does each Canadian province have a College of Pharmacists? Is this a statutory body? What is their role? Who is represented in Canadian Colleges of Pharmacists? Only pharmacists (community sector only, or also hospital sector?; if community sector, only pharmacy owners or also employed pharmacists?), or are further pharmacy professionals also involved in a College of Pharmacists? I am asking because in the paper pharmacy technicians are mentioned. What is the role of pharmacy technicians in New Brunswick and, in general, in Canada? (not particularly linked to POCT – as across the globe, the mandate of pharmacy technicians varies).

Abstract: The New Brunswick College of Pharmacists (NBCP) Strategic Plan 2023 is mentioned, with a quote (“support the evolving role and well-being of pharmacy professionals”). A quote appears to be too specific for an abstract.

In addition, the relationship between this quote (a strategic objective?) and the POCT is not clear. Also, from a public health perspective, I would argue that it is important to improve patient and population health. Though I see the importance of “well-being of pharmacy professional”, I would not stress it, in particular not with regard to POCT.

Abstract: “environmental scan of the ten Canadian provincial regulatory authorities” – what is meant? In addition, from the main body of the text (reference 30), I learn that apparently the authors have not done this scan, but the Canadian HTA body did it. Thus, the authors reported CADTH’s findings (again, the missing methods section leads to a situation where I had to take some assumptions).

Overall, the abstract was written in a very broad manner and lacked focus. In a revision of the paper, the abstract also requires major rephrasing, and the research question needs to be clarified.

Introduction, line 39: acronym “ASSURED”: why does the quote relate to a systematic literature review, and not the World Health Organization, which introduced this concept? In addition, meanwhile, the concept has been further developed toward REASSURED.

Introduction, line 44: “recent” – the authors should avoid writing “recent”, if this paper is aimed to offer value also in 10-20 years

Introduction, line 45: it is not clear what the authors relate to when they talk about “outcomes-based”

Introduction, second paragraph, presentation of new regulations: the language is very technical and Canadian-specific, it is difficult to understand for non-Canadian readers. The “minimum expectations” were not clear; are minimum regulatory requirements meant?

Line 63: “It is important to analyze the various POCTs that pharmacy professionals utilize in other regions to understand how New Brunswick pharmacy professionals may implement them into their practice and the impact on patient care.“ This sentence should not be part of the results (I consider chapters 2 and the following chapters as results chapter, even if not officially called so), but this would rather be the rationale for the paper.

Line 72: “Pharmacists are medication therapy experts.” Though I agree, this is very much marketing language. Kindly aim to avoid such phrases.

Line 101: A study on an alert system for antibiotics was reported. However, this information was provided in the section on chronic diseases.

Line 189: “Quality assurance or control is a term used to ensure the performance of devices is acceptable.” Please make sure to specify that this definition of “quality assurance” is used for the purpose of this paper, but “quality assurance” usually has a much broader meaning.

Line 224: PRAs – please explain the abbreviation (provincial regulatory authority?)

Line 239: PEI – please explain the abbreviation

Line 274: information on pharmacies with a waiver in the US. First, this is old information (as of 2015), and it would gain more value if not the absolute numbers were reported but a percentage

Line 278: “prescriptive authority and action” – jargon language – what is meant?

Line 279/280: “In recent regulation changes, states such as Idaho allow independent prescribing based on a POCT result.” First, see the previous comment on “recent”. Out of the context, I do not understand. “Independent prescribing” – does it mean prescribing by a pharmacist? Why independent? Without control mechanism? Is this allowed? Prescribing of which medicines?

Line 288 / paragraph: it is not clear which POCT are addressed

Line 333 ff – chapter 6 on pharmacy technicians’ role in POCT: Following up on what I said earlier, the role of pharmacy technicians in POCT has to be put into the context of what pharmacy technicians are permitted in different countries. The authors quote literature but when it checked the references 56-58, references 57-58 related to the US, and reference 56 was a review (geographic scope was not clear). If these pieces of literature are kept in a revised version, the different country context needs to be made clear to the readers.

Author Response

The paper is called “Establishing a Point-of-Care Testing Policy: The New Brunswick College of Pharmacists Experience”.

I expected to learn about the experience in implementing the point-of-care test (POCT) policy in New Brunswick: what are its features, why was it introduced, and which policy objectives should be reached (and which KPIs were defined for measuring progress), which difficulties in preparation and implementation were encountered, and which were the experiences of pharmacists, whether or not, and to which extent, the objectives were reached, possible hindering and enabling factors.

- The title and abstract have been edited to set more appropriate expectations regarding the paper per your feedback. The paper intends to describe the process we took to inform the policy as differences exist in how jurisdictions approach policy writing. Regulatory authorities with a policy in place may utilize this information when their policy is under review. Additionally, the information contained in the paper may be beneficial to jurisdictions considering implementing POCT services. Finally, the information can extend beyond regulatory bodies to pharmacy professionals seeking to implement POCT. Currently, the policy has been approved by NBCP Council. NBCP is working towards implementing the policy. Our next steps are to determine key performance indicators that may evaluate the utility of the policy in practice. This description has been added to Section 10. 

The abstract already suggested that “the New Brunswick College of Pharmacists Experience”, as stated in the title, might not be addressed. In the Introduction, recent regulatory amendments as of June 2022 were mentioned. I was wondering how an impact assessment were possible, after such a short time. But I was still expecting to learn about the preparation and implementation phase. But apart from a brief outlook in the Introduction, only section 8 informed about the policy development.

- The title has been edited to notify the reader that the paper focuses on the process of drafting the policy rather than a focus on implementing.

In the abstract, the New Brunswick College of Pharmacists (NBCP) Strategic Plan 2023 is mentioned. I was confused because this Strategic Plan points to the future, and this might not be the basis for the implementation of a measure whose impact were intended to evaluated.

- Details regarding the Strategic Plan have been removed.

The abstract states: “With the approval of the amendments, NBCP developed a policy to guide pharmacy professionals in implementing safe POCT services that provide quality patient care. As part of NBCP’s policy development process, a literature search and environmental scan of the ten Canadian provincial regulatory authorities along with other countries was completed.” I had to continue reading subsequent sections of the paper to understand that findings of this review were content in this paper. I will come back and comment on the objective of the paper below.

- The abstract has been revised to specify that the findings are contained within the paper.  

In the Introduction, second paragraph, the new regulations are presented, with an outlook on the roles of the different professionals in (community) pharmacy. Then it is written: “With minimum expectations set in place, the NBCP sought to consider aspects of POCT that would help to set parameters to support patient safety. This paper will explore the usage of POCT in community pharmacies, its impact on patient outcomes, and quality control measures.” I was wondering how the report of the implementation of the POCT policy in New Brunswick is already possible a few weeks after the regulatory change. The following sentence (“An analysis of provincial regulatory authorities, other countries, education perspectives, and the role of pharmacy technicians will provide an understanding of the POCT environment in pharmacy practice to inform policy development”) and the rest of the paper taught me that the measure is, at the time of the writing of the paper, being implemented, and no findings on experiences or learnings is to be expected.

- Agree that the expectations were unclear. The edits in the introduction align with the fact that the paper focuses on the steps taken to draft the policy and outlines that the implementation stage is being finalized.

Information on the methods is missing (apart from mentioning of the literature review and “environmental scan” in the abstract – see also below specific comments) though it might have helped to understand. This paper would benefit from a methods section. The remainder of the paper is the presentation of a targeted review on some elements of POCT in community pharmacy globally and in Canada. However, it is not clear why certain topics were included and others not. Given the missing methods section, the readers do not know why some pieces of literature were included, and others were not. It rather reads as a technical report, or textbook, with some “self-marketing” for the pharmacists’ profession (see below).

- We have added a methods section that describes the literature review and environmental scan that was undertaken. 

I do not say that the information reported is not interesting. But I do not understand why this paper was written. If it aims to inform colleagues in the NBCP, other formats than a scientific paper (e.g. a booklet) might be more appropriate. There is no need to do a narrative review of systematic reviews in a paper.

I urge for a complete rewrite of the paper, upon clarification of its aim. Any misleading information on a possible reporting of experiences of the new policy should be avoided; thus the title needs to be changed. The authors need to define a research question. They can report some of the findings of the review, but they need to present the findings in light of the new POCT policy in New Brunswick: What does the international experience tell them? How did it feed their policy development? What are learnings to consider?

Specific comments:

The “New Brunswick College of Pharmacists” (NBCP): For readers not familiar with the Canadian pharmacy system, it would be helpful to provide further information on the NBCP. Does each Canadian province have a College of Pharmacists? Is this a statutory body? What is their role? Who is represented in Canadian Colleges of Pharmacists? Only pharmacists (community sector only, or also hospital sector?; if community sector, only pharmacy owners or also employed pharmacists?), or are further pharmacy professionals also involved in a College of Pharmacists? I am asking because in the paper pharmacy technicians are mentioned. What is the role of pharmacy technicians in New Brunswick and, in general, in Canada? (not particularly linked to POCT – as across the globe, the mandate of pharmacy technicians varies).

- Thank you for highlighting this gap in information. A description of pharmacists being self-regulated in each province of Canada is described in the second paragraph of the Introduction section. Section 8 has additional information added regarding pharmacy technicians responsibility.

Abstract: The New Brunswick College of Pharmacists (NBCP) Strategic Plan 2023 is mentioned, with a quote (“support the evolving role and well-being of pharmacy professionals”). A quote appears to be too specific for an abstract. In addition, the relationship between this quote (a strategic objective?) and the POCT is not clear. Also, from a public health perspective, I would argue that it is important to improve patient and population health. Though I see the importance of “well-being of pharmacy professional”, I would not stress it, in particular not with regard to POCT.

- The quote has been removed from the abstract following a rework of this section. 

Abstract: “environmental scan of the ten Canadian provincial regulatory authorities” – what is meant? In addition, from the main body of the text (reference 30), I learn that apparently the authors have not done this scan, but the Canadian HTA body did it. Thus, the authors reported CADTH’s findings (again, the missing methods section leads to a situation where I had to take some assumptions).

- The methods section should clear this discrepancy. The CADTH environmental scan was an article found through the literature review. The environmental scan was a separate task NBCP completed that involved investigating the other nine provincial regulations and policies surrounding POCT along with other countries that have POCT guidelines such as the US, Australia, and Ireland.

Overall, the abstract was written in a very broad manner and lacked focus. In a revision of the paper, the abstract also requires major rephrasing, and the research question needs to be clarified.

Introduction, line 39: acronym “ASSURED”: why does the quote relate to a systematic literature review, and not the World Health Organization, which introduced this concept? In addition, meanwhile, the concept has been further developed toward REASSURED.

- This concept has been updated in the introduction.

Introduction, line 44: “recent” – the authors should avoid writing “recent”, if this paper is aimed to offer value also in 10-20 years

- Agree, this has been removed.

Introduction, line 45: it is not clear what the authors relate to when they talk about “outcomes-based”

- Described concepts that were added to the regulations “including details regarding collaboration with other providers, referring where appropriate, and requirement of documentation.”

Introduction, second paragraph, presentation of new regulations: the language is very technical and Canadian-specific, it is difficult to understand for non-Canadian readers. The “minimum expectations” were not clear; are minimum regulatory requirements meant?

- Edited this statement to make it more clear that the NAPRA standards would need to be met in addition to the policy put forth by the College.

Line 63: “It is important to analyze the various POCTs that pharmacy professionals utilize in other regions to understand how New Brunswick pharmacy professionals may implement them into their practice and the impact on patient care.“ This sentence should not be part of the results (I consider chapters 2 and the following chapters as results chapter, even if not officially called so), but this would rather be the rationale for the paper.

- This line has been moved to the introduction.

Line 72: “Pharmacists are medication therapy experts.” Though I agree, this is very much marketing language. Kindly aim to avoid such phrases.

- This line has been removed.

Line 101: A study on an alert system for antibiotics was reported. However, this information was provided in the section on chronic diseases.

- The alerts were notifying the pharmacist regarding kidney function for antibiotics that required dose adjustments based on eGFR. Due to the study performing POCT to determine the patients kidney function in order to take action on antibiotic dosing, it was included in the Chronic Disease section. This section has been edited to iterate that the test result is renal function.

Line 189: “Quality assurance or control is a term used to ensure the performance of devices is acceptable.” Please make sure to specify that this definition of “quality assurance” is used for the purpose of this paper, but “quality assurance” usually has a much broader meaning.

- The definition has been updated to add “in the context of POCT”.

Line 224: PRAs – please explain the abbreviation (provincial regulatory authority?)

- Authors have defined this term Provincial Regulatory Authority (PRA) in Section 2.2.

Line 239: PEI – please explain the abbreviation (added abbreviation)

- Prince Edward Island (a Canadian province) has been edited within the text.

Line 274: information on pharmacies with a waiver in the US. First, this is old information (as of 2015), and it would gain more value if not the absolute numbers were reported but a percentage

- This absolute number may no longer be relevant therefore it has been removed.  

Line 278: “prescriptive authority and action” – jargon language – what is meant?

- Changed to pharmacist prescribing

Line 279/280: “In recent regulation changes, states such as Idaho allow independent prescribing based on a POCT result.” First, see the previous comment on “recent”. Out of the context, I do not understand. “Independent prescribing” – does it mean prescribing by a pharmacist? Why independent? Without control mechanism? Is this allowed? Prescribing of which medicines?

- Removed “recent”. Clarified that it is pharmacist prescribing for influenza and streptococcus.

Line 288 / paragraph: it is not clear which POCT are addressed

- Reworded sentence. POCT generally as it would be limited to approved devices in each country.

Line 333 – chapter 6 on pharmacy technicians’ role in POCT: Following up on what I said earlier, the role of pharmacy technicians in POCT has to be put into the context of what pharmacy technicians are permitted in different countries. The authors quote literature but when it checked the references 56-58, references 57-58 related to the US, and reference 56 was a review (geographic scope was not clear). If these pieces of literature are kept in a revised version, the different country context needs to be made clear to the readers.

- The literature on pharmacy technicians came out of the US particularly Tennessee and Ohio which has been clarified. These two states require technician certification. The role in these states relates to that of pharmacy technicians scope in New Brunswick.

Reviewer 3 Report

Authors Hutchings and Shiamptanis describe the use of POCT in a pharmacy setting, quality assurance, regulatory framework, educational opportunities, and the role of pharmacy technicians in relation to POCT. Although many aspects are covered and in part, authors do talk about the limitation of POCT by pharmacists, I think it is to include a subsection, esp in 2.2 Infectious Disease POCT on how doctor-patient confidentiality will be addressed or should be addressed.  

Author Response

Thank you for your feedback. The same confidentially standards that apply to pharmacy practice would apply to POCT services. Details regarding patient confidentiality and privacy are outlined within the NAPRA Standards and would apply in the context of POCT services. Confidentiality details were added to the introduction when the NAPRA Standards are mentioned. 

Round 2

Reviewer 2 Report

Thank you for revising the article. With the description of the distinction between the process of the work on writing the POCT policy and the findings from literature and other surveys the paper has gained considerable clarity.

The paper would, however, still benefit if you could strengthen the outline. Now, apart from the methods, there are always one-digit headings, and after presentation of findings, section 9 and section 10 relate again on the New Brunswick POCT policy. For the readers, it would be easier if you could reorganise the narrative (maybe using more 2-digit headings).

Author Response

The paper would, however, still benefit if you could strengthen the outline. Now, apart from the methods, there are always one-digit headings, and after presentation of findings, section 9 and section 10 relate again on the New Brunswick POCT policy. For the readers, it would be easier if you could reorganise the narrative (maybe using more 2-digit headings).

The following sections were combined under common headers; POCT across & outside of Canada, education & pharmacy technician, and engagement & policy. Grouping the sections should improve the flow.